chemical biology/biochemistry

activation mechanism, Bruton's tyrosine kinase, T474M mutation

**Author for correspondence:**
Quhuan Li
e-mail: liqh@scut.edu.cn

†Present address: School of Bioscience and Bioengineering, South China University of Technology, Guangzhou 510006, Guangdong, People's Republic of China.

This article has been edited by the Royal Society of Chemistry, including the commissioning, peer review process and editorial aspects up to the point of acceptance.

# A mechanism for localized dynamics-driven activation in Bruton's tyrosine kinase

Simei Qiu[1,2], Yunfeng Liu[1] and Quhuan Li[1,2,†]

[1]Institute of Biomechanics/School of Bioscience and Bioengineering, and [2]Guangdong Provincial Engineering and Technology Research Center of Biopharmaceuticals, South China University of Technology, Guangzhou People's Republic of China

QL, 0000-0001-6133-3114

Bruton's tyrosine kinase (BTK) plays a vital role in mature B-cell proliferation, development and function. Its inhibitors have gradually been applied for the treatment of many B-cell malignancies. However, because of treatment-associated drug resistance or low efficacy, it is urgent to develop new inhibitors and/or improve the efficacy of current inhibitors, where finding the intrinsic activation mechanism becomes the key to solve this problem. Here, we used BTK T474M mutation as a resistance model for inhibitors to study the mechanism of BTK activation and drug resistance by free molecular dynamics simulations. The results showed that the increase of kinase activity of T474M mutation is coming from the conformation change of the activation ring and ATP binding sites located in BTK N-terminus region. Specifically, the Thr[474] mutation changed the structure of A-loop and stabilized the binding site of ATP, thus promoting the catalytic ability in the kinase domain. This localized dynamics-driven activation mechanism and resistance mechanism of BTK may provide new ideas for drug development in B-cell malignancies.

## 1. Introduction

Many protein kinases are considered key molecules in cancer, and they constitute major drug targets since certain kinase signalling pathways are functionally essential for many types of cancer [1]. To date, the US Food and Drug Administration (FDA) has approved many kinase inhibitors for the treatment of multiple solid and haematological tumours [2,3]. Bruton's tyrosine kinase (BTK), a cytoplasmic tyrosine kinase, is the only member of the Tec family of kinases that has been related to the pathogenesis of cancer in humans [4]. Ibrutinib, a first-generation inhibitor of BTK, was approved in 2013 and has shown noteworthy clinical activity in lymphoid malignancies, such as chronic lymphocytic leukaemia (CLL) [5] and mantle cell lymphoma (MCL) [6]. More selective drugs, including

calabrutinib [7] and zanubrutinib [8], were later developed as second-line inhibitors for the treatment of B-cell malignancies.

In normal B cells, BTK can be activated by a large array of receptors and signalling pathways, including the B-cell receptor, the Toll-like receptor, various chemokine receptors, and Fc receptor signalling. It then triggers a series of signalling events resulting in transcriptional regulation of gene expression, increased calcium mobilization and flux, cytoskeletal rearrangements, and other cellular responses [9]. BTK vitally affects many B-cell functions, including their reproduction, survival and homing, and inherited mutations that render BTK dysfunctional can result in X-linked agammaglobulinaemia [10]. Since BTK is a key molecule for B-cell biology, BTK inhibitors have been developed and applied to treat some B-cell malignancies. Most kinase inhibitors in clinical use are reversible ATP-competitive inhibitors [11]. Similarly, most BTK inhibitors also act on the kinase domain of BTK, recognizing unique features of the ATP binding pocket. For example, ibrutinib, a first-generation inhibitor, interacts with residue Cys[481] on the ATP binding site. However, a major obstacle in targeted kinase inhibitor therapy is drug resistance, generally due to point mutations, low selectivity, and off-target activity, or due to other effects on the targeted kinases. Therefore, new inhibitors that deal better with these problems are urgently needed. The strategies used earlier to develop kinase inhibitors were based on static chemical concepts, such as electrophilicity [12]. However, biological processes are dynamic and intimately connected with structural changes in the molecules involved. Although structural changes are key elements for our understanding of kinase activity, their precise nature remains unclear. Furthermore, the emergence of mutations requires the exploration of additional key amino acids as drug binding sites. Similarly, understanding the allosteric mechanisms regulating the kinase domain would offer valuable ideas to search for key sites, and promote the research and development of inhibitors in BTK-dependent B-cell malignancies.

Gatekeeper residues are present in many kinases and play a very important role. For example, gatekeeper residues regulate the phosphorylation activity and nuclear entry of the ERK kinase [13], and mutations of such residues often cause resistance to ERK kinase inhibitors [14]. Some researchers also found that in BTK, residue Thr[474] acts as a gatekeeper, playing a vital role, since when it is mutated into methionine, the activity of the kinase domain is greatly increased [15–17]. The same mutation had a negative impact on the function of several covalent BTK inhibitors and non-covalent substrate analogues, indicating that it can cause some BTK inhibitors to lose potency. At the same time, it could amplify the effects of other mutations, such as E513G [17]. Many researchers have used the Thr[474] mutation to test the efficacy of new inhibitors [16,18]. However, although current studies show that the Thr[474] mutation promotes BTK kinase activity, it is not clear how it activates the kinase and how resistant it is. Therefore, we conducted computer simulations to explore the activation mechanism of the kinase domain and analyse the structural changes caused by the T474M mutation to gain a deeper understanding of the activation mechanism of BTK and provide ideas for drug development in B-cell malignancies.

# 2. Material and methods

## 2.1. System set-up

Molecular dynamics (MD) simulations were performed with four structures: the wild-type (WT) kinase domain of BTK (Protein Data Bank (PDB) accession number 6E4F), the T474M mutant kinase domain, the WT kinase domain in complex with ATP and the T474M mutant kinase domain in complex with ATP. The structure of the mutant T474M kinase domain was modelled by mutating Thr[474] into Met[474] in the structure of the WT kinase domain using the 'mutate residue' plugin in the visual molecular dynamics (VMD 1.9.2) software. The WT-ATP complex was built by combining the structure of the WT kinase domain of BTK with that of ATP extracted from another protein kinase complex (PDB accession number 3A7H), using the AUTOPSF plugin in VMD. The structure of the T474M-ATP complex was built in a similar way. Then, VMD was used to lead the protein structures in a rectangular water box and add certain ions to neutralize the system and maintain ATP (figure 1c)

## 2.2. MD simulations

Two main software packages were used for the simulations: the VMD software was used for visualization and modelling [19] and the NAMD 2.13 program was used for the MD simulations [20].

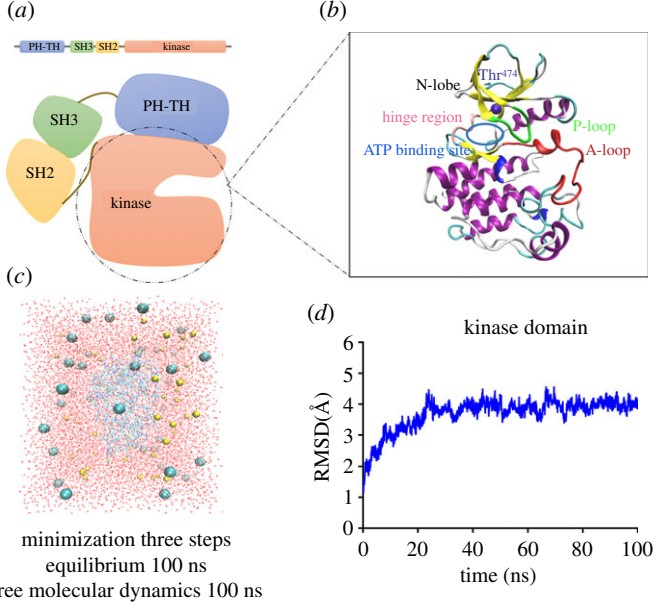

**Figure 1.** The structure of BTK and the equilibrium process of MD simulations. (a) Structural model of full-length BTK. (b) Structural model for the catalytic kinase domain of BTK. (c) System set-up of MD simulations. The yellow spheres represent Na$^+$ ions, the blue spheres represent Cl$^-$ ions, and the red points represent water molecules. (d) Root mean square deviation (RMSD) versus time profile of the global kinase domain (residues 389–658), which shows that simulation system has reach a relatively stable state after 100 ns of equilibrium.

For two structures that do not contain ATP, the force field CHARMM27 and the particle mesh Ewald summation method for the calculation of electrostatic interactions with a 12 Å cut-off length were used to perform MD simulations for all atoms, using a 2.0 fs step and periodic boundary conditions [21,22]. The two systems, WT and T474M, were subjected to energy minimizations according to a protocol allowing all protein atoms to be fixed during the first 15 000 minimization steps. Then, the heavy atoms of the protein were fixed for the following 15 000 minimization steps, and another 15 000 steps followed with free atoms. The water molecules were free in each run. The temperature of the minimized systems increased gradually from 0 to 310 K. Each system was equilibrated three times for 100 ns (figure 1d) and under controlled conditions of pressure (1 atm) and temperature (310 K). For each system, the final equilibrium structure was chosen as the initial conformation for free MD simulations, in order to reduce contingencies in protein structure selection. The free MD simulations were run once for over 100 ns for each structure. The trajectories were recorded and analysed using VMD. For two structures with ATP, we used the force field CHARMM36. All other operations were the same as described above.

## 2.3. Data analysis

All structures and results were visualized and analysed using VMD tools. The root mean square deviation (RMSD), radius of gyration ($R_{gyr}$) and Cα root mean square fluctuation (RMSF) values for the protein backbone were calculated to investigate conformational changes and stability in the structures. The RMSF values could also be used to mark local structural flexibilities. The process by which the DFG motif moved closer to β6β7 in the kinase domain [23] was quantified by the angle of two straight lines, one formed between Leu[518] in β6 and Asn[526] in β7, and the other between Leu[518] and Asp[539] in the DFG motif. A hydrogen bond was defined when the donor–acceptor distance was less than 0.35 nm and the donor–hydrogen-acceptor angle was less than 30°. The distance between the DFG motif and the ATP binding site was calculated considering the centre points of the two structures. Calculation of solvent-accessible surface area (SASA) values, using a probe with a radius of 0.14 nm, was used to assess residue exposure to the solvent. A salt bridge was defined when the distance between the nitrogen atoms of basic residues and the oxygen atoms of acidic residues was less than 0.4 nm. The hydrogen bond and salt bridge occupancy were calculated by the proportion of bond survival time during the simulation period.

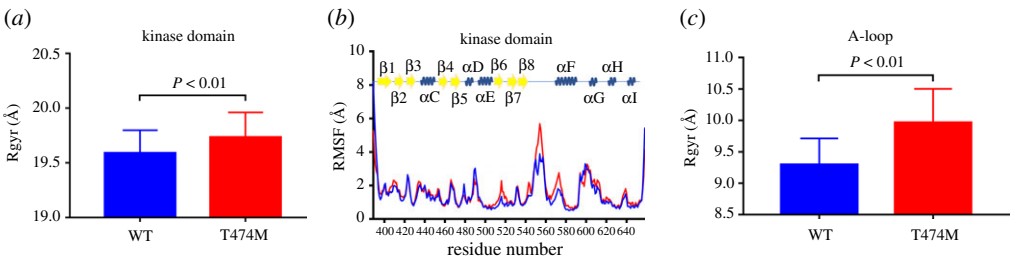

**Figure 2.** Global stability and flexibility of the WT (blue) and T474M mutant (red) kinase domain. (a) $R_{gyr}$ of the global kinase domain. (b) Distribution of RMSF (WT, blue; T474M, red) on the residue chains of the kinase domain. (c) $R_{gyr}$ of the A-loop (residues 539–559). Total data from three independent runs are shown. The secondary structure assignments are based on information from the PDB archives, and the data presented are averages over all three runs. Sheets are shown in yellow and helices in blue. The $p$-values show statistically significant differences.

## 2.4. Statistical analysis

The statistical difference between groups was assessed by the unpaired two-tailed Student's $t$-test. When the $p$-value was less than 0.05, the difference was considered statistically significant.

## 3. Results

### 3.1. The T474M mutation induces a conformation transition in the kinase domain

In order to illuminate the structural foundation explaining why the mutation of Thr[474] enhances BTK enzyme activity [15], we conducted free MD simulations on the structures of the WT and T474M kinase domains, and recorded the values of $R_{gyr}$, expressing the backbone radius of gyration, and RMSF, marking the C$\alpha$ root mean square fluctuations. The simulations were performed for 100 ns with a step of 2 fs and were repeated three times for each structure. We surmised that a mutation-induced change in the structure of the global kinase domain (residues 389–658) occurred, based on the time evolution of the backbone $R_{gyr}$ (figure 2a) values. We observed that the mutation had remarkably increased $R_{gyr}$, indicating that the mutant increases the flexibility of the structure. We calculated the C$\alpha$ RMSF value for each residue in the kinase domain and found that the mutation promoted skeleton mobility (figure 2b), with the most obvious fluctuation occurring at the C-terminus of the activation loop (A-loop) (residues 550–558). We wondered if the A-loop (residues 539–559) caused the significant overall change in $R_{gyr}$, thus, we analysed the $R_{gyr}$ of the A-loop alone. Indeed, we found an increase in the $R_{gyr}$ of the A-loop in the mutant compared to the WT kinase domain (approx. 0.6 Å; figure 2c), remarkably reflecting the internal structural change of the mutant. These structural changes were also directly visible in a superimposition of the two structures (figure 3). The structural heterogeneity is illustrated in the two parts of the A-loop: the N-terminal containing the DFG motif (figure 3, upper box), and the C-terminal containing Tyr[551] (figure 3, lower box). The activation ring, which is closely related to the activation of kinases in general, is stronger in BTK than in the other Tec family kinases [24], which may be responsible for the large number of signalling pathways activated by BTK and its more important role in human physiology. Our results show that the T474M mutation triggers a change in the kinase domain, mediating a conformation transition, mainly by increasing the flexibility of the A-loop, which further affects BTK activation.

### 3.2. The T474M mutation triggers a movement bringing the DFG motif of the A-loop near the ATP binding site and exposing the C-terminal of the A-loop

Our results show that the major difference between the WT and T474M structures is in the A-loop (figure 2). We speculated that the mutation induced a decrease in the distance between the DFG motif and the ATP binding site and exposed Tyr[551] (figure 3). To confirm the reduction of the distance between the DFG and ATP binding site, we measured (i) the angle between the two straight lines formed between Leu[518] in $\beta$6 and Asn[526] in $\beta$7 and between Leu[518] and Asp[539] in DFG (figure 4c), and (ii) the distance between the DFG motif and the ATP binding site (figure 4b), measured between the DFG motif and the centre between the P-loop and the hinge region (figure 4f). To verify the Tyr[551]

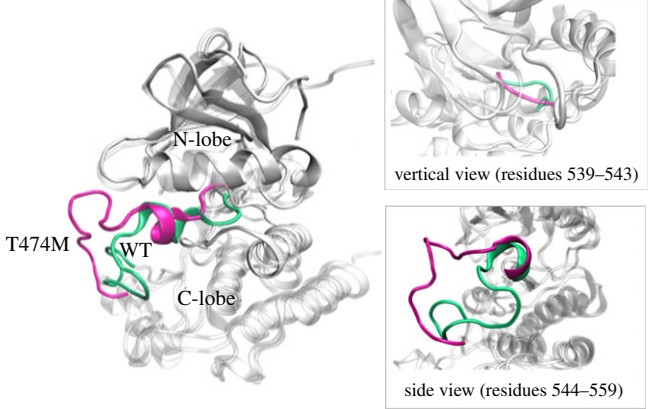

**Figure 3.** The mutation-triggered local change in the structure of the WT kinase domain. Superposition of the conformations of WT (magenta) and T474M (green) kinase domains. Each structure was randomly selected from one frame. The two boxes on the right show a magnification of the A-loop. The upper box presents a vertical view, showing the first part of the A-loop (residues 539–543), which includes the DFG motif (residues 539–541); the lower box presents a side view, showing the sequence from Arg[544] until Phe[559], which contains Tyr[551]. The upper box shows that in the mutant, the DFG motif is closer to the loop containing Asn[526] and the ATP binding site. The lower box shows the increased exposure to water of some residues in the mutant compared to the WT kinase domain.

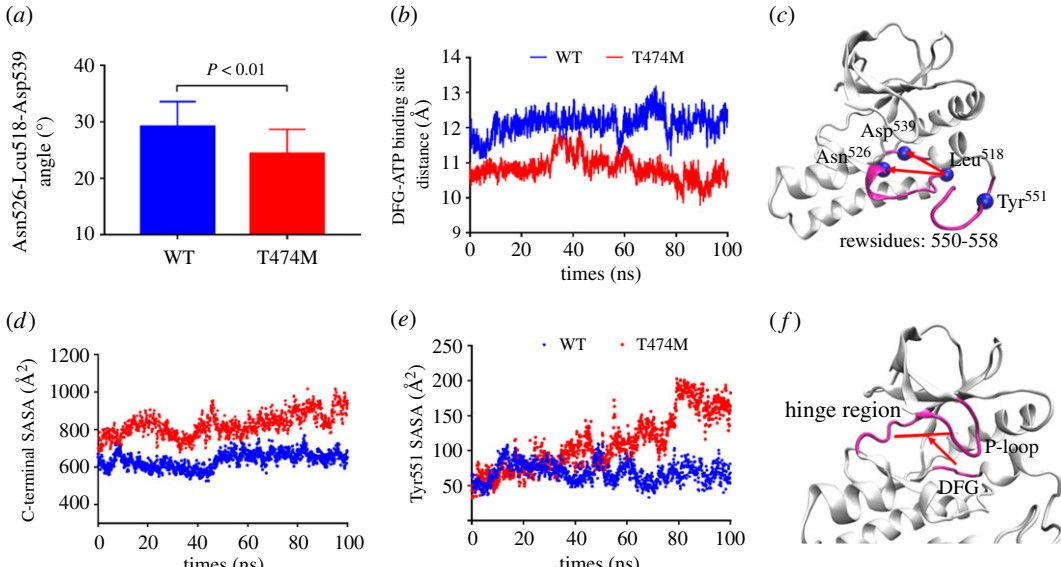

**Figure 4.** The distance between the DFG motif and the ATP binding site and the SASA values of the C-terminal part of the A-loop. (a) Conformational changes bringing the DFG motif closer to $\beta6\beta7$ containing Asn[526], which was quantified by the angle ($\theta$) between two straight lines, one formed between Leu[518] in $\beta6$ and Asn[526] in $\beta7$, and the other between Leu[518] and Asp[539] in the DFG motif. (b) The distance between the centre of the DFG motif and the ATP binding site. (c) The presentation of angles in the structure. (d) SASA values of residues Glu[550]-Lys[558]. (e) SASA values of Tyr[551]. (f) The presentation of distances in the structure. Each of the time–distance profiles presents an average of results based on three independent runs. The p-value indicates a statistically significant difference.

exposure, we calculated the average SASA value of the hydrophobic residues 550–558 (figure 4d) and of Tyr[551] (figure 4e). The results show a decrease of approximately 4° in the above-mentioned angle in the mutant compared to the WT structure (figure 4a), indicating that in the mutant, the DFG motif is closer to $\beta7$. The directly measured distance between the DFG motif and the ATP binding site was also shorter in the T474M mutant than in the WT kinase domain, further indicating that the DFG motif in the T474M mutant is indeed nearer to the ATP binding site. These changes in angle and distance reflect the two ways in which the DFG motif can move closer to the space of the ATP binding site. The mutation decreases the distance between the DFG motif and the ATP binding site, facilitating the chelation of

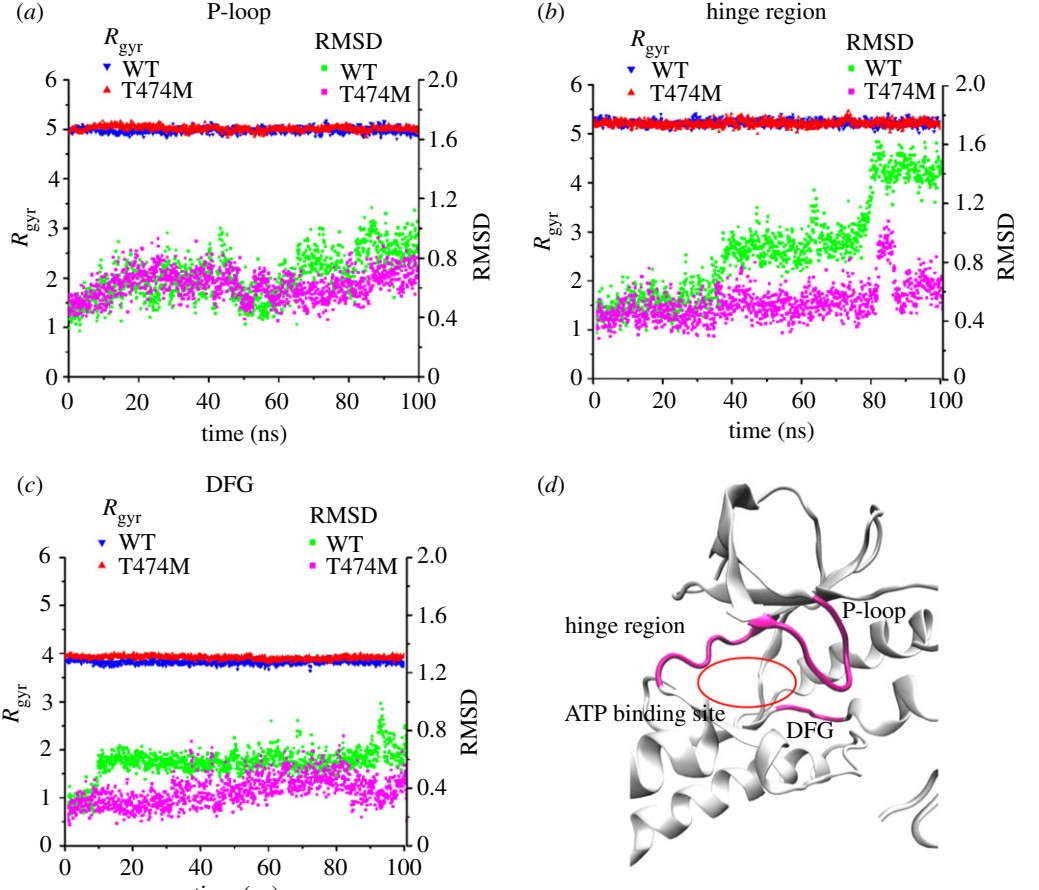

**Figure 5.** Stabilization of the ATP binding site in the T474M mutant. (*a*) Time–$R_{gyr}$ profiles on the left side of the P-loop (residues 410–415), and time–RMSD profiles on the right side of the loop. (*b*) Similar profiles as in (*a*), with respect to the hinge region (residues 475–479). (*c*) Similar profiles as in (*a*), with respect to the DFG motif (residues 539–541). The data shown represent averages from three runs. The RMSD values at the right side of all three domains are lower and with less fluctuations in the mutant than in the WT kinase domain. (*d*) Cartoon representation of the crystal structure of the kinase domain, with the hinge region, P-loop and DFG motif shown in magenta, and the ATP binding site shown in red.

the aspartate residue of the DFG motif with $Mg^{2+}$ ions and the formation of an ATP-$Mg^{2+}$ complex, which is vital and necessary for the reaction with ATP. It is well known that $Tyr^{551}$ phosphorylation is the first step required for kinase activation. Therefore, $Tyr^{551}$ exposure may accelerate BTK activation. We found a marked increase in the SASA values of residues 550–558 and $Tyr^{551}$ in the mutant in comparison with the WT (figure 4*d,e*), indicating that the mutation of $Thr^{474}$ induces $Tyr^{551}$ exposure. Another consequence of this mutation is that the C-terminal region of the A-loop moves further away from the protein centre and becomes more exposed to water, driving structural changes in other parts of the protein, similar to the structural changes that activate the kinase domain after phosphorylation of $Tyr^{551}$ in the WT molecules. In summary, the mutation decreases the distance between the DFG motif and the ATP binding site and exposes $Tyr^{551}$, promoting the opening of the A-loop structure, and consequently, BTK activation.

## 3.3. The T474M mutation stabilizes the ATP binding site

The T474M mutation can greatly increase BTK kinase activity [15–17], but little is known about the structural changes it induces in the ATP binding site. For this, we analysed the structure of the ATP binding site. In particular, we examined the $R_{gyr}$ and RMSD values of the P-loop (figure 5*a*), hinge region (figure 5*b*) and DFG motif (figure 5*c*). It is well known that the N-lobe of the kinase domain contains the P-loop (residues 410–415), which helps localizing the phosphate groups of ATP, and the hinge region (residues 475–479), which helps positioning the ATP adenosine. On the other hand, $Asp^{539}$ in the DFG motif at the ATP binding site chelates $Mg^{2+}$ ions. These three structural

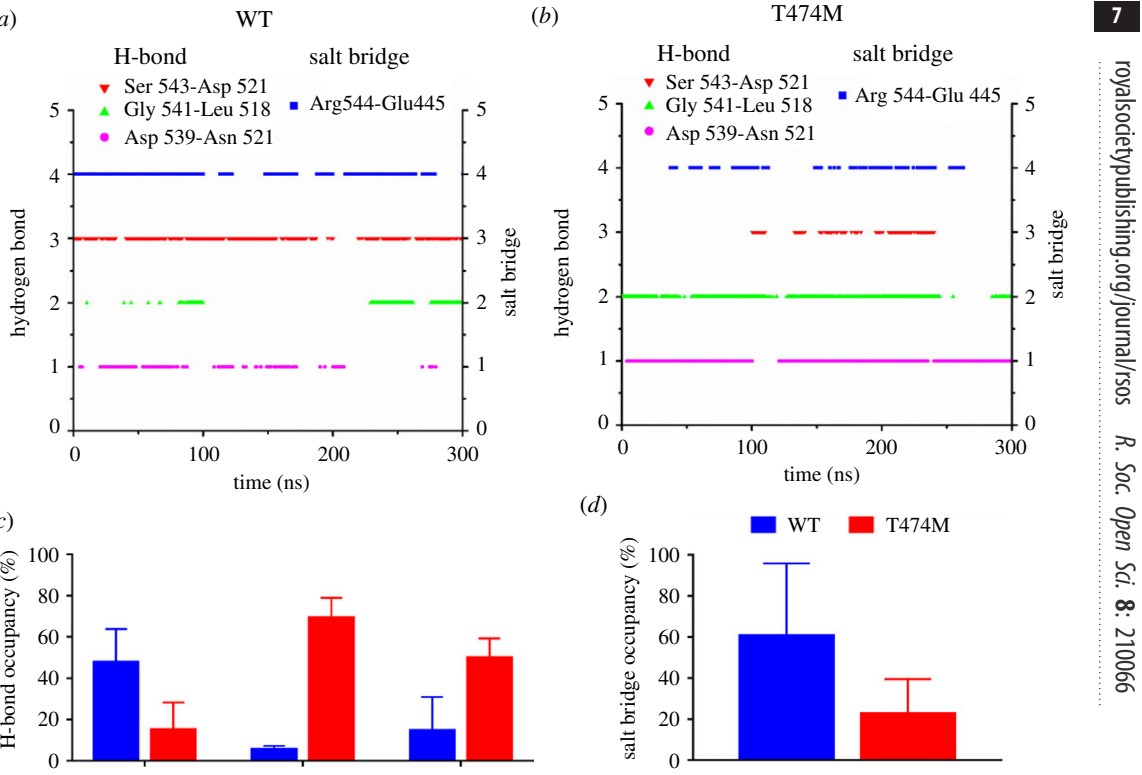

**Figure 6.** Key hydrogen bonds and salt bridges in the A-loop of the kinase domain. (*a*) Survival of the Ser[543]-Asp[521], Gly[541]-Leu[518] and Asp[539]-Asn[526] hydrogen bonds (left), and of the Arg[544]-Glu[445] salt bridge (right) in the WT kinase domain. (*b*) Same as in (*a*) for the T474M mutant. (*c*) Survival rate of the three hydrogen bonds in the WT and the T474M mutant kinase domains. (*d*) Survival rate of the salt bridge (Arg[544]-Glu[445]). The data shown in (*a*) and (*b*) are results from all three independent runs. The data shown in (*c*) and (*d*) represent averages from three independent runs, with mean and standard deviation (s.d.) values shown.

components are related to ATP binding, stability and reaction, and our results (figure 5*a–c*) show that they become more stable in the T474M mutant, as indicated by lower RMSD values at the right side of the three parts and similar $R_{gyr}$ values at the left side, compared with the WT structure. This suggests an increased stability of the ATP binding site (figure 5*d*), securing the presence of ATP. In order to verify this hypothesis, we docked ATP into the WT and T474M kinase domain structures, and performed free MD simulations on the structures of the resulting complexes, following the same method used for the free structures. We found a higher number of hydrogen bonds in the mutant than in the WT complex, indicating that ATP binding was indeed more stable in the mutant structure (electronic supplementary material, figure S1A). The stable presence of ATP in the T474M mutant may provide sufficient time and opportunity for the ATP reaction to occur.

## 3.4. The activation mechanism of the BTK catalytic domain is mediated by four key interactions, as indicated by the WT and T474M structures

The T474M mutation changed the local dynamic properties of the A-loop structure. However, it is unclear whether these A-loop structural changes are related to dynamic changes in the network of salt bridges and hydrogen bonds. Therefore, we examined the related inner salt bridges and hydrogen bonds of the A-loop by free MD simulations and found four key interactions: one salt bridge (Arg[544]-Glu[445]) and three hydrogen bonds (Ser[543]-Asp[521], Gly[541]-Leu[518] and Asp[539]-Asn[526]) (figure 6*a,b*). Leu[518] and Asn[526] are located at β6 and β7, respectively. Asp[539], Gly[541] and Arg[544] are located at the A-loop, and Glu[445] is located at the αC-helix. Arg[544] formed a firm salt bridge with Glu[445] (occupancy 60%) and Ser[543] formed a relatively firm hydrogen bond with Asp[521] (occupancy 48%) in the WT kinase domain (figure 6*a,c,d*). These interactions were weak (less than occupancy 20%) in the T474M mutant (figure 6*b–d*). On the contrary, the other two hydrogen bonds (Gly[541]-Leu[518] and Asp[539]-Asn[526]) were weak (less than occupancy 20%) in the WT protein (figure 6*a,c*) and strong (greater than

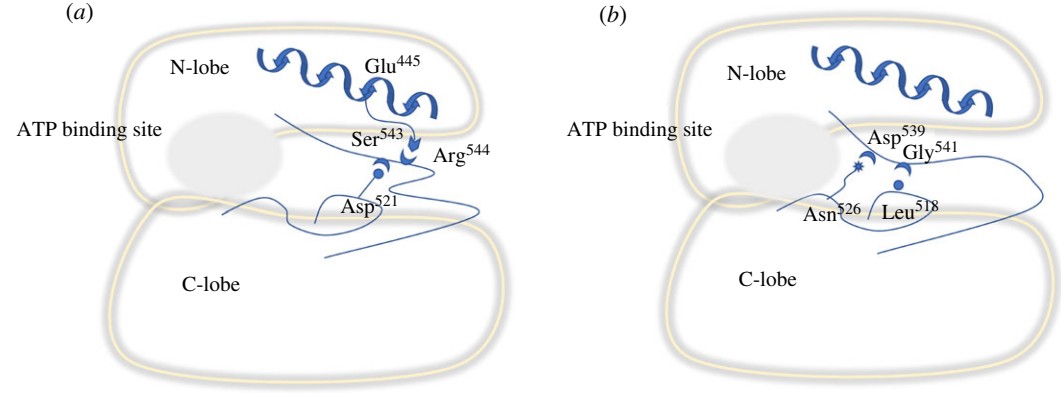

**Figure 7.** Activation mechanism of the BTK catalytic domain based on the WT and T474M mutant structures. Cartoon representation of the structural pathway mediating BTK activation in the T474M mutant. (*a*) In the WT kinase domain, Arg[544] and Ser[543] in the A-loop interact with Glu[445] in the $\alpha$C-helix and Asp[521] in $\beta$6, respectively. (*b*) In the T474M mutant, the two pairs of interactions are destroyed and the C-terminal regions of the A-loop becomes exposed to water, leading to the exposure of Tyr[551]. Two additional hydrogen bonds are formed (Gly[541]-Leu[518] and Asp[539]-Asn[526]) were formed, allowing the DFG motif of the A-loop to come closer to the ATP binding site, facilitating catalysis.

occupancy 50%) in the T474M mutant (figure 6*b,c*). Thus, the T474M mutation appears to weaken the firm salt bridge between Arg[544] and Glu[445] and the relatively firm hydrogen bond between Ser[543] and Asp[521], while strengthening the weak hydrogen bonds between Gly[541] and Leu[518] and between Asp[539] and Asn[526] (figure 7). The analysis of the A-loop hydrogen bonds and salt bridges shows that the bond survival rates of the Ser[543]-Asp[521] hydrogen bond and the Arg[544]-Glu[445] salt bridge were 30–40% lower in the mutant than in the WT kinase domain, possibly rendering the C-terminal region of the A-loop more flexible, facilitating its exposure to the solvent. Similarly, the survival rates of the Gly[541]-Leu[518] and Asp[539]-Asn[526] hydrogen bonds were higher in the mutant than in the WT kinase domain, especially that of the Gly[541]-Leu[518] hydrogen bond, which was nearly 70% higher. These results indicate that this pair of hydrogen bonds played a crucial role in positioning the A-loop of the DFG motif closer to the ATP binding site. All four pairs of hydrogen bonds are likely to cooperate with each other and change the structure of the A-loop (figure 7), thus, affecting kinase activity.

# 4. Discussion

B-cell tumours are haematologic tumours characterized by long drug dependence, and it is very important to study drugs for B-cell tumour treatment, especially those acting on BTK. Currently, the T474M mutation is prominent in the study of BTK inhibitors [16,17,25], and the structural mechanism underlying the function of this mutation has also been explored to some extent [15]. However, it has been explored only at the experimental level, and the activation mechanism of the BTK kinase domain at atomic level remains unclear. Therefore, this study aimed at providing insights on the allosteric activation mechanism of the BTK kinase domain by the T474M mutation, in order to dictate new directions for the research and development of new inhibitors for B-cell malignancies. Moreover, since gatekeeper mutations appear in many kinases [14,26], our study can also provide very effective information and ideas for drug research on other kinases.

Similar to other kinases, the catalytic domain of BTK contains an N-lobe and a C-lobe (figure 1*b*), and ATP and Mg$^{2+}$ ions bind in the hinge region connecting the two lobes. The activation ring (A-loop) in the C-lobe has two functions: the phosphorylation of C-terminal Tyr[551] that greatly increases BTK activity, and the chelation of Mg$^{2+}$ ions through its N-terminal DFG motif (a highly conserved aspartate-phenylalanine-glycine motif). In the N-lobe, the glycine-rich phosphate localization ring (P-loop) helps positioning the phosphate group of ATP. In this study, we found that mutation of Thr[474] greatly changed the structural flexibility of the A-loop (figure 2). On the one hand, its N-terminal DFG sequence became closer to the ATP binding site, and on the other hand, its C-terminal region (residues 550–558) became exposed to water (figure 4). These transformations were accompanied by changes in the interaction between four pairs of residues (figure 6), namely, the Arg[544]-Glu[445] salt bridge and the Ser[543]-Asp[521], Gly[541]-Leu[518] and Asp[539]-Asn[526] hydrogen bonds. In the T474M mutant,

the Arg$^{544}$-Glu$^{445}$ and Ser$^{543}$-Asp$^{521}$ residue pairs were disconnected from each other, while the Gly$^{541}$-Leu$^{518}$ and Asp$^{539}$-Asn$^{526}$ interaction were strengthened. This may be a result of an outward movement of the C-terminal end of the A-loop, accompanied by an inward movement of its N-terminal end, bringing the DFG sequence closer to the ATP binding site and stabilizing ATP, thus, setting the stage for ATP reaction. At the same time, the T474M mutation stabilizes the movements of the ATP binding site (figure 5). These structural changes are probably responsible for the significant increase in kinase activity by improving ATP binding and reaction.

The aspartate of the DFG motif of the A-loop can modulate drug binding because it is protonated [27]. Different protonation states are related with different types of inhibitors [28], and the role of DFG in BTK has been already highlighted by several researchers. In our study, the process by which the DFG motif approaches the ATP binding site may be coordinated and regulated by three pairs of interactions: Ser$^{543}$-Asp$^{521}$, Gly$^{541}$-Leu$^{518}$ and Asp$^{539}$-Asn$^{526}$. In the mutant, the disconnection of Ser$^{543}$ from Asp$^{521}$ provides conditions enhancing the flexibility of DFG, while the strengthening of the Gly$^{541}$-Leu$^{518}$ and Asp$^{539}$-Asn$^{526}$ interactions stabilizes the DFG motif closer to the ATP binding site. DFG-Asp$^{539}$ can then chelate with Mg$^{2+}$ ions at the ATP binding site and form an ATP-Mg$^{2+}$ complex, which is required for the kinase reaction to occur. In other words, T474M brings DFG closer to the ATP binding site, providing conditions favourable for ATP stability and reaction in BTK. Our results confirm the stability of ATP in the mutant (electronic supplementary material, figure S1A) and the proximity of the DFG motif to ATP (electronic supplementary material, figure S1B). Therefore, these three pairs of interactions may be the basis of kinase activation in the T474M mutant, even during the normal activation process.

It is well known that the hinge region of the BTK kinase domain is bound to the adenosine ring of ATP, while the P-loop binds to the phosphate groups of ATP. These two sites are relatively stable in the WT kinase domain (figure 5a,b), whereas ATP is stable in the T474M mutant (electronic supplementary material, figure S1A), indicating that the mutation is likely to promote stable ATP binding and kinase activity. In addition, we found that three residues, namely Lys$^{430}$, Arg$^{525}$ and Lys$^{558}$, were associated with the stable presence of ATP in the mutant (electronic supplementary material, figure S1C-D), and they may be important for kinase activity. In summary, the T474M mutation provides a suitable and relatively stable space for ATP binding and reaction, thereby accelerating the reaction and improving response efficiency.

The T474M mutation greatly increases the flexibility and structural fluctuations of the C-terminal part of the A-loop (residues 550–558), which is accompanied by the short bond survival time of the salt bridge between Arg$^{544}$ and Glu$^{445}$. The Arg$^{544}$-Glu$^{445}$ salt bridge is very important for the activation state of BTK because the first step during BTK activation is Tyr$^{551}$ phosphorylation, following by the destruction of the Arg$^{544}$-Glu$^{445}$ salt bridge, which allows the formation of the Glu$^{445}$-Lys$^{430}$ salt bridge, leading to BTK activation [29]. We hypothesize that the T474M mutation may be similar to the activation mechanism in the kinase domain after phosphorylation at site Tyr$^{551}$ of the normal structure, which will help us to better understand the activation process of BTK.

In addition, we compared two loss-of-function mutations, G414R and G525Q, which were reported in X-linked agammaglobulinaemia (XLA) diseases [30,31], with the gain-of-function mutation T474M. Structural changes caused by T474M did not appear in the loss-of-function mutations. Such mutations have low flexibility, even lower than WT. This was confirmed regardless of the conformation changes in the A-loop, stability of ATP binding sites, and survival of four interacting bond pairs (data not provided).

There are five structural domains in BTK (figure 1a), each of which has its own activation mechanism, and then jointly form the overall activation process of BTK. To date, many studies have explored to some extent the activation mechanism of different BTK domains [15,23,32–37]. For example, some studies have shown PH dimerization [34,38], which could mediate the dimerization of the catalytic region of BTK [23], and that the A-loop may be associated with kinase domain dimerization [39]. Although this study explored the activation mechanism of the kinase domain, the activation mechanism of the entire structure of BTK is still not clear, requiring further studies.

Currently, simulations are used to investigate the binding of inhibitors to the kinase domain [28], targeting only specific inhibitors. Only by truly understanding the allosteric mechanisms in the activation of the BTK kinase domain can the research, development and improvement of inhibitors be better served. Therefore, understanding the normal activation process of BTK will provide many valuable ideas for the research of BTK inhibitors. In addition, current studies have shown that BTK inhibitors can be used to treat solid tumours, such as colon cancer [40], breast cancer [41] and gastric carcinoma [42], by promoting tumour cell apoptosis. Therefore, they may have a profound effect on the treatment of other tumours as well.

# 5. Conclusion

In this work, we found the mutation at site $Thr^{474}$ greatly changed the A-loop of BTK. On the one hand, the N-terminal DFG sequence of A-loop was close to the ATP binding site to set the stage for ATP reaction, and on the other hand, the C-terminal (residues 550–558) was exposed to water to facilitate phosphorylation at $Tyr^{551}$. These transformations are accompanied by a change in the interaction of four groups, one salt bridge $Arg^{544}$-$Glu^{445}$ and three hydrogen bonds $Ser^{543}$-$Asp^{521}$/$Gly^{541}$-$Leu^{518}$/$Asp^{539}$-$Asn^{526}$. These structural changes are probably responsible for mechanism for localized dynamics-driven activation and resistance in BTK, and these four pairs of key interaction residues can be used as targets for new drugs. In conclusion, our study may provide new ideas for new drug creation and development in treating B-cell malignancies.

Data accessibility. Data available from the Dryad Digital Repository: https://doi.org/10.5061/dryad.mgqnk98zb.

Authors' contributions. Q.L. and S.Q. were responsible for the overall design and investigation. S.Q. and Y.L. performed data analysis. S.Q. and Q.L. were responsible for manuscript writing and revising. All authors participated in discussion of the results.

Competing interests. We declare we have no competing interests.

Funding. This work was supported by National Natural Science Foundation of China grant no. 31870928 (Q.L.) and the Natural Science Foundation of Guangdong Province, China no. 2021A1515010040 (Q.L.).

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
