## [Peer Review File · Royal Society Open Science]

Review History

RSOS-210066.R0 (Original submission)

Review form: Reviewer 1

Is the manuscript scientifically sound in its present form?

No

Are the interpretations and conclusions justified by the results?

No

Is the language acceptable?

Yes

Do you have any ethical concerns with this paper?

No

Have you any concerns about statistical analyses in this paper?

No

Recommendation?

Major revision is needed (please make suggestions in comments)

Comments to the Author(s)

he manuscript needs certain improvements.

- The chosen mutation T474M is quite significant, especially considering the fact that by introducing such a mutation, a complete different electron scheme or better said the electronic cloud is changed significantly, thus all the subsequent differences between the WT and T474M can be expected.
- Moreover by changing a tyrosine with methionine also a sulfuric atom is introduced instead of an aromatic system originating from tyrosine, meaning that the $\pi - \pi$, electrostatic, and all other interactions are also significantly changed.
- Also mutations with other amino acid residues should be made, and then the results of molecular simulations should be compared between each other. They should be also compared with the experimental results.
- Since the word is about a plausible mechanism, it would be beneficial to do also the QM/MM studies, and not only MD studies.

Review form: Reviewer 2

Is the manuscript scientifically sound in its present form?

No

Are the interpretations and conclusions justified by the results?

No

Is the language acceptable?

Yes

Do you have any ethical concerns with this paper?

No

Have you any concerns about statistical analyses in this paper?

No

Recommendation?

Major revision is needed (please make suggestions in comments)

Comments to the Author(s)

The authors have used standard molecular dynamics to compare the structure of wild-type BTK with that of a mutant, without and with ATP. While the study comes across as a bit dated in terms of methodology, I think the differences in structure revealed by the MD are likely valid and they are interesting. I think the manuscript could meet RSOS standards of scientific soundness if the authors revise it to meet the following points of critique:

- 1) Although "straight MD" is less popular than methods employing enhanced sampling, I think that, within the limitations of the force-field, the trajectories have been long enough (100 ns or 70 ns) to reveal interesting and important structural changes upon mutation.. (no action required for this point).
- 2) My main critique is with the authors (mis) use of the term "stability". Mutation does not lead to instability. It leads to a different, but stable structure. Indeed, MD is only valid if sampling is sufficient for stable thermodynamic properties. I think the whole manuscript has to be revised to avoid this kind of misconception. For example, on p.4., "mutant is internally unstable" is

incorrect; the mutant would have been unstable if the structure hadn't changed, but the new (equilibrated) structure is stable.

3) In the title, I don't understand "localized dynamics-driven activation". I did understand from the manuscript that there are important structural changes, but what is "dynamics-driven"?

4) Please justify the use of the dated CHARMM22 parameters/

5) What is the justification for different simulation times (100ns for WT, 70 for mutant)

6) On p. 3, NAMD has nothing to do with nonadiabatic (non-Born-Oppenheimer) dynamics.

Originally the program was named "not another MD" program. It is sometimes these days known as Nanoscale MD, but the "official" reference just avoids the issue, using NAMD as a label without definition (<https://aip.scitation.org/doi/10.1063/5.0014475>)

Decision letter (RSOS-210066.R0)

Dear Dr Li:

Title: A Mechanism for Localized Dynamics-driven Activation in BTK

Manuscript ID: RSOS-210066

The editor assigned to your manuscript has now received comments from reviewers. We would like you to revise your paper in accordance with the referee and Subject Editor suggestions which can be found below (not including confidential reports to the Editor). Please note this decision does not guarantee eventual acceptance.

Please submit your revised paper before 18-Jun-2021. Please note that the revision deadline will expire at 00.00am on this date. If we do not hear from you within this time then it will be assumed that the paper has been withdrawn. In exceptional circumstances, extensions may be possible if agreed with the Editorial Office in advance. We do not allow multiple rounds of revision so we urge you to make every effort to fully address all of the comments at this stage. If deemed necessary by the Editors, your manuscript will be sent back to one or more of the original reviewers for assessment. If the original reviewers are not available we may invite new reviewers.

On behalf of the Subject Editor Professor Anthony Stace and the Associate Editor Dr Andrew Harned.

RSC Associate Editor:

Comments to the Author:

The reviewers have expressed some interest in the work reported by this manuscript, even if the methods used are bordering on routine. I agree, that the conclusions could prove useful to others. The referees also raise several important points that would strengthen the manuscript. The authors should carefully consider these concerns and submit a revised paper accordingly.

RSC Subject Editor:

Comments to the Author:

(There are no comments.)

Reviewers' Comments to Author:

Reviewer: 1

Comments to the Author(s)

he manuscript needs certain improvements.

- The chosen mutation T474M is quite significant, especially considering the fact that by introducing such a mutation, a complete different electron scheme or better said the electronic cloud is changed significantly, thus all the subsequent differences between the WT and T474M can be expected.
- Moreover by changing a tyrosine with methionine also a sulfuric atom is introduced instead of an aromatic system originating from tyrosine, meaning that the $\pi - \pi$, electrostatic, and all other interactions are also significantly changed.
- Also mutations with other amino acid residues should be made, and then the results of molecular simulations should be compared between each other. They should be also compared with the experimental results.
- Since the word is about a plausible mechanism, it would be beneficial to do also the QM/MM studies, and not only MD studies.

Reviewer: 2

Comments to the Author(s)

The authors have used standard molecular dynamics to compare the structure of wild-type BTK with that of a mutant, without and with ATP. While the study comes across as a bit dated in terms of methodology, I think the differences in structure revealed by the MD are likely valid and they are interesting. I think the manuscript could meet RSOS standards of scientific soundness if the authors revise it to meet the following points of critique:

- 1) Although "straight MD" is less popular than methods employing enhanced sampling, I think that, within the limitations of the force-field, the trajectories have been long enough (100 ns or 70 ns) to reveal interesting and important structural changes upon mutation.. (no action required for this point).
- 2) My main critique is with the authors (mis) use of the term "stability". Mutation does not lead to instability. It leads to a different, but stable structure. Indeed, MD is only valid if sampling is sufficient for stable thermodynamic properties. I think the whole manuscript has to be revised to avoid this kind of misconception. For example, on p.4., "mutant is internally unstable" is incorrect; the mutant would have been unstable if the structure hadn't changed, but the new (equilibrated) structure is stable.
- 3) In the title, I don't understand " localized dynamics-driven activation". I did understand from the manuscript that there are important structural changes, but what is "dynamics-driven"?
- 4) Please justify the use of the dated CHARMM22 parameters/
- 5) What is the justification for different simulation times (100ns for WT, 70 for mutant)
- 6) On p. 3, NAMD has nothing to do with nonadiabatic (non-Born-Oppenheimer) dynamics. Originally the program was named "not another MD" program. It is sometimes these days known as Nanoscale MD, but the "official" reference just avoids the issue, using NAMD as a label without definition (<https://aip.scitation.org/doi/10.1063/5.0014475>)

Author's Response to Decision Letter for (RSOS-210066.R0)

See Appendix A.

Decision letter (RSOS-210066.R1)

Dear Dr Li:

Title: A Mechanism for Localized Dynamics-driven Activation in BTK
Manuscript ID: RSOS-210066.R1

It is a pleasure to accept your manuscript in its current form for publication in Royal Society Open Science. The chemistry content of Royal Society Open Science is published in collaboration with the Royal Society of Chemistry.

On behalf of the Subject Editor Professor Anthony Stace and the Associate Editor Dr Andrew Harned.

RSC Associate Editor
Comments to the Author:
The authors appear to have addressed the comments and concerns raised in the previous review.
I can now recommend publication.

Reviewer(s)' Comments to Author:

Appendix A

Dear Editors and Reviewers:

Thank you for your letter and for the reviewers' comments concerning our manuscript entitled "A Mechanism for Localized Dynamics-driven Activation in BTK" (ID: RSOS-210066). Those comments are valuable and very helpful for revising and improving our paper. We have studied comments carefully and have made some corrections which we hope meet with approval. Revised portions are marked in yellow in the manuscript. The responses to the reviewer's comments are presented following.

We would love to thank you for allowing us to resubmit a revised copy of the manuscript and we highly appreciate your time and consideration.

Best regard!

Quhuan Li

June 17, 2021

Reviewer #1:

Q1: The chosen mutation T474M is quite significant, especially considering the fact that by introducing such a mutation, a complete different electron scheme or better said the electronic cloud is changed significantly, thus all the subsequent differences between the WT and T474M can be expected.

Response : Thank you for your summary and the affirmation of my work.

Q2: More over by changing a tyrosine with methionine also a sulfuric atom is introduced instead of an aromatic system originating from tyrosine, meaning that the π - π , electrostatic, and all other interactions are also significantly changed.

Response : Thank you for pointing out other points on which we can focus. In the present study, we converted threonine to methionine, which mainly introduced the sulfuric atom.

Q3: Also mutations with other amino acid residues should be made, and then the results of molecular simulations should be compared between each other. They should be also compared with the experimental results.

Response: To compare other mutations, we also used G414R and G525Q, two loss-of-function mutations that lead to BTK inactivation, which were reported to be found in XLA diseases^{1, 2}. Compared to the gain-of-function T474M mutation studied in our research, the two loss-of-function mutations were generally similar to WT, thus resembling locking in an inactivated state; this was confirmed regardless of the changes in the A-loop, the stability of ATP binding sites,

or the survival of four interacting bond pairs. The following is a comparison between the two.

- ① Regarding overall changes in internal flexibility, we found that the R_{gyr} of the loss-of-function mutations was lower than that of both WT and gain-of-function mutations (Fig. 1A), indicating that the kinase domains of loss-of-function mutations were compact. Such tightness may substantially reduce the probability of ATP binding. In contrast, the gain-of-function mutation in the kinase domain is more flexible and is primarily reflected in the A-loop (Fig. 1B), which favors ATP binding.

Figure 1. Global stability and flexibility of the WT, G414R, R525Q, and T474M mutants. (A) R_{gyr} of the global kinase domain. (B) Distribution of RMSF on the residue chains of the kinase domain; WT is shown in blue, G414R in red, R525Q in green, and T474M in purple. The P -value indicates a statistically significant difference.

- ② Regarding changes in the activation of A-loop, its DFG region was far from the ATP binding site for loss-of-function mutations. This is primarily reflected by the larger angle (Fig. 2A) and distance (Fig. 2B). Moreover, from the C-terminus of the A-loop, the exposure was not as evident as that of T474M (Fig. 2C, D). These results showed that loss-of-function mutations maintain a locking A-loop, resulting in the inactivation of BTK.

Figure 2. The distance between the DFG motif and ATP binding site, and the SASA values of the C-terminal part of the A-loop. (A) Conformational changes of A-loop. The angle (θ)

made by three residues Leu⁵¹⁸, Asn⁵²⁶, and Asp⁵³⁹. (B) Distance between the center of the DFG motif and the ATP binding site. (C) SASA values of residues Glu⁵⁵⁰-Lys⁵⁵⁸. (D) SASA values of Tyr⁵⁵¹. Each of the time-distance profiles presents an average of the results based on three independent runs. The *P*-value indicates a statistically significant difference.

- ③ The RMSD values of the P-loop, DFG, and Hinger regions of loss-of-function mutations were measured (Fig. 3) to evaluate the stability of the ATP binding sites. The fluctuation of loss-of-function mutations was larger than that of the gain-of-function T474M mutation, indicating that ATP could not be maintained at the binding site for loss-of-function mutations because of the large fluctuations.

Figure 3. Stabilization of the ATP binding site. Time- R_{gyr} profiles on the left side of the P-loop (residues 410–415) (A), the DFG motif (residues 539–541) (B), and the hinge region (residues 475–479) (C).

- ④ Regarding the key hydrogen bond and salt bridge in A-loop, the formation and breakage of hydrogen bonds and salt bridges in loss-of-function mutations were similar to those in WT, but not in the gain-of-function mutation T474M (Fig. 4). This indicates that the variation of the A-loop is caused by the mutation at Thr474 and further illustrates the importance of these four pairs of interactions on kinase activity.

Figure 4. Key salt bridges and hydrogen bonds in the A-loop of the kinase domain. The survival of Arg⁵⁴⁴-Glu⁴⁴⁵ (blue) salt bridge and three pair of hydrogen bonds (Ser⁵⁴³-Asp⁵²¹ (red), Gly⁵⁴¹-Leu⁵¹⁸ (green), and Asp⁵³⁹-Asn⁵²⁶ (purple)) in WT (A), T474M mutant (B),

G414R mutant (C), and R525Q mutant (D).

- ⑤ Regarding experimental verification, we did not conduct wet experiments in our study; however, there are some reports on the kinase activity of the gain-of-function mutation T474M in the existing literature. These studies showed that the mutation T474M promoted kinase activity primarily through the transition of the conformation of activation loops in BTK. The related evidence list in the following literature is summarized as follows: Compared with WT type, the enzyme activity of BTK in the T474M mutation significantly increased^{3, 4} (Fig. 5A, B) by increasing the phosphorylation level of BTK⁵ (Fig. 5C), which then promoted the tolerance to some BTK⁴ inhibitors (Fig. 5D) and enhanced BTK activity. In addition, studies have shown that the BTK activation ring is important for the function of BTK and other kinases, such as ITK⁶. These results demonstrate that the mutation changes the activation loop conformation and then affects the activity of BTK, as determined in this study.

Figure 5. Effect of mutations on enzyme activity. (A) Kinase activities in T474M. (B) Steady state kinetic constants for in BTK between WT and mutants. (C) Immunoblot analysis of BTK Y223 autophosphorylation. (D) Biochemical potency of inhibitors against WT and mutants.

Q4: Since the word is about a plausible mechanism, it would be beneficial to do also the QM/MM studies, and not only MD studies.

Response: Thank you for recommending QM/MM for our study. In our study, we performed MD for WT and mutation BTK to reveal the related activity mechanism, consistent with the experiment. Many simulations using MD methods have been reported^{7,8}. Therefore, we believe that MD is reliable for this type of research.

Reviewer #2:

The authors have used standard molecular dynamics to compare the structure of wild-type BTK with that of a mutant, without and with ATP. While the study comes across as a bit dated in terms of methodology, I think the differences in structure revealed by the MD are likely valid and they are interesting. I think the manuscript could meet RSOS standards of scientific soundness if the authors revise it to meet the following points of critique:

Q1: Although "straight MD" is less popular than methods employing enhanced sampling, I think that, within the limitations of the force-field, the trajectories have been long enough (100 ns or 70 ns) to reveal interesting and important structural changes upon mutation. (no action required for this point).

Response: We appreciate the reviewer's positive evaluation of our work.

Q2: My main critique is with the authors (mis) use of the term "stability". Mutation does not lead to instability. It leads to a different, but stable structure. Indeed, MD is only valid if sampling is sufficient for stable thermodynamic properties. I think the whole manuscript has to be revised to avoid this kind of misconception. For example, on p.4., "mutant is internally unstable" is incorrect; the mutant would have been unstable if the structure hadn't changed, but the new (equilibrated) structure is stable.

Response: Thank you for your comment. We have modified this expression throughout the text accordingly. We have modified the phrase "unstable state" to the more rigorous statement, "conformation transition." Details of the modifications can be found in Results 4.1 section.

Q3: In the title, I don't understand " localized dynamics-driven activation". I did understand from the manuscript that there are important structural changes, but what is "dynamics-driven"?

Response: The explanation for "localized dynamics-driven activation" is as follows: Localized refers to the activation ring region. Dynamics-driven activation indicates that the T474M mutation triggers a conformational change in the activation ring that promotes kinase activity. This conformational activation ring change is a localized force-driven process because of the interaction change in the mutation. Only the transformation of the activation ring can change the BTK domain in the active state.

Q4: Please justify the use of the dated CHARMM22 parameters/

Response: The force field used in this study was CHARMM27, as confirmed after double checking; the necessary modifications have been made in the corresponding text of the manuscript. In addition, it should also be noted that after adding ATP, the CHARMM36 force field was used because CHARMM27 does not contain ATP parameters; therefore, this section is also supplemented in the Methods section of the text.

Q5: What is the justification for different simulation times (100ns for WT, 70 for mutant)

Response: We used a simulation time of 70 ns for the system containing ATP in our study. As

the results tended to be stable at 70 ns of simulation time, we were able to conclude that the BTK domain in the T474M mutation had a more stable internal structure when bound with ATP. The corresponding parameters included more hydrogen bonds (Fig. 6A), a shorter centroid distance between DFG and ATP (Fig. 6B), and higher survival rates of key hydrogen bonds (Fig. 6C, D) in mutant T474M than in WT. To maintain consistency with the primary content, we extended the simulation time to 100 ns, with the new results agreeing with the original ones.

Figure 6. ATP binding is more stable owing to the mutant T474M. (A) The number of hydrogen bonds with time in wild-type kinase domain and mutant T474M; (B) Centroid distance between DFG and ATP; (C) Survival rates of key hydrogen bonds between BTK domain and ATP in WT and mutant T474M; and (D) Schematic of three pairs of hydrogen bonds interacting with ATP with high survival rates in T474M, and not in WT. Magenta represents the ATP.

Q6: On p. 3, NAMD has nothing to do with nonadiabatic (non-Born-Oppenheimer) dynamics. Originally the program was named "not another MD" program. It is sometimes these days known as Nanoscale MD, but the "official" reference just avoids the issue, using NAMD as a label without definition (<https://aip.scitation.org/doi/10.1063/5.0014475>)

Response: Thanks for the related information you gave us here.

Finally, thank you for your careful review. We really appreciate your efforts in reviewing our manuscript during this unprecedented and challenging time. We wish good health to you, your family, and community. Your careful review has helped to make our study clearer and more comprehensive.

Reference

1. Vihinen M, V. D., Maniar HS, Ochs HD, Zhu Q, Vorechovský I, Webster AD, Notarangelo LD, Nilsson L, Sowadski JM., Structural basis for chromosome X-linked agammaglobulinemia: a tyrosine kinase disease. *Proc Natl Acad Sci U S A*, **1994**, *91* (26), 12803-7.
2. Holinski-Feder E, W. M., Brandau O, Jedele KB, Nore B, Bäckesjö CM, Vihinen M, Hubbard SR, Belohradsky BH, Smith CI, Meindl A., Mutation screening of the BTK gene in 56 families with X-linked agammaglobulinemia (XLA): 47 unique mutations without correlation to clinical course. *Pediatrics* **1998**, *101* (2), 276-284.
3. Joseph, R. E.; Xie, Q.; Andreotti, A. H., Identification of an allosteric signaling network within Tec family kinases. *J Mol Biol* **2010**, *403* (2), 231-42.
4. Johnson, A. R.; Kohli, P. B.; Katewa, A.; Gogol, E.; Belmont, L. D.; Choy, R.; Penuel, E.; Burton, L.; Eigenbrot, C.; Yu, C.; Ortwine, D. F.; Bowman, K.; Franke, Y.; Tam, C.; Estevez, A.; Mortara, K.; Wu, J.; Li, H.; Lin, M.; Bergeron, P.; Crawford, J. J.; Young, W. B., Battling BTK Mutants With Noncovalent Inhibitors That Overcome Cys481 and Thr474 Mutations. *ACS Chem Biol* **2016**, *11* (10), 2897-2907.
5. Wang, S.; Mondal, S.; Zhao, C.; Berishaj, M.; Ghanakota, P.; Batlevi, C. L.; Dogan, A.; Seshan, V. E.; Abel, R.; Green, M. R.; Younes, A.; Wendel, H. G., Noncovalent inhibitors reveal BTK gatekeeper and auto-inhibitory residues that control its transforming activity. *JCI Insight* **2019**, *4* (12).
6. Joseph, R. E.; Kleino, I.; Wales, T. E.; Xie, Q.; Fulton, D. B.; Engen, J. R.; Berg, L. J.; Andreotti, A. H., Activation loop dynamics determine the different catalytic efficiencies of B cell- and T cell-specific tec kinases. *Sci Signal* **2013**, *6* (290), ra76.
7. Wang, Q.; Pechersky, Y.; Sagawa, S.; Pan, A. C.; Shaw, D. E., Structural mechanism for Bruton's tyrosine kinase activation at the cell membrane. *Proc Natl Acad Sci U S A* **2019**, *116* (19), 9390-9399.
8. Zhang, Y.; Lin, Z.; Fang, Y.; Wu, J., Prediction of Catch-Slip Bond Transition of Kindlin2/ β 3 Integrin via Steered Molecular Dynamics Simulation. *Journal of Chemical Information and Modeling* **2020**, *60* (10), 5132-5141.